# Sustainable Particleboards Based on Brewer’s Spent Grains

**DOI:** 10.3390/polym16010059

**Published:** 2023-12-23

**Authors:** Lucia Rossi, Lucia Wechsler, Mercedes A. Peltzer, Emiliano M. Ciannamea, Roxana A. Ruseckaite, Pablo M. Stefani

**Affiliations:** 1Instituto de Investigaciones en Ciencia y Tecnología de Materiales (INTEMA), Consejo Nacional de Investigaciones Científicas y Técnicas (CONICET), Universidad Nacional de Mar del Plata (UNMdP), Av. Colón 10850, Mar del Plata B7600FDQ, Argentina; luciarossig@gmail.com (L.R.); luciawec@gmail.com (L.W.); emiliano@fi.mdp.edu.ar (E.M.C.); roxana@fi.mdp.edu.ar (R.A.R.); 2Departamento de Ciencia y Tecnología, Consejo Nacional de Investigaciones Científicas y Técnicas (CONICET), Universidad Nacional de Quilmes, Roque Sáenz Peña 352, Bernal B1876BXD, Argentina; mercedes.peltzer@unq.edu.ar

**Keywords:** brewer’s spent grain, binderless board, mechanical properties, physical properties, self-adhesion

## Abstract

Brewer’s spent grain (BSG) is the main solid waste generated in beer production and primarily consists of barley malt husks. Based on the active promotion of circular economy practices aimed at recycling food industry by-products, this study assessed for the first time the production of particleboards based on BSG as the sole source of lignocellulosic material and natural adhesive without the use of additives or other substrates. In order to achieve particleboards from entirely sustainable sources, BSG particles have to self-bind by thermo-compression with water. In this context, the aim of this study is to assess the effects of pressing temperatures and particle size on properties such as modulus of elasticity, modulus of rupture, internal bond, thickness swelling, and water absorption. The performance of binderless boards was compared with that of a control panel (control) using BSG combined with phenolic resin. Processing conditions were selected to produce boards with a target density of 1000 kg/m³ and a thickness of 5 mm. To confirm the efficiency of the self-adhesion process, scanning electron microscopy was used to examine the boards. The processes of self-adhesion and particle-to-particle contact were facilitated at a pressing temperature of 170 °C and a particle size range of 200–2380 µm (ground BSG), resulting in improved flexural properties and enhanced water resistance. The properties of BSG-based binderless boards were comparable to those reported for other biomass residues, suggesting that they might be used in non-structural applications, such as interior decoration.

## 1. Introduction

Particleboards, a fundamental component within the global wood-based panel industry, play an essential role in meeting the escalating demand for engineered wood products worldwide. The production of wood-based particleboard currently reaches about 105 million m^3^ per year [1] and is in continuous growth. The growing demand for raw materials, predominantly wood fibers, veneers, and particles, is putting substantial pressure on native and cultivated forests, leading to deforestation, ecosystem disruption, and biodiversity loss. As a result, the wood industry is challenged to explore sustainable sourcing strategies, alternative raw materials, and recycling initiatives to alleviate the stress on forestry resources to promote a more environmentally responsible production approach [2,3,4,5,6,7]. Vitrone et al. [8] and Pędzik et al. [9] have reported on reviews that focus on the economic and sustainability aspects of different alternative lignocellulosic resources for particleboard production. Their analyses cover diverse fiber sources, including agricultural residues and forestry waste. Both studies underscore the significance of decreasing wood consumption by substituting with lignocellulosic residues, thereby contributing to mitigating climate change and aligning with the circular economy’s “close the loop” concept. Several agro-industrial and agroforest residues have been explored for particleboard manufacturing, such as bagasse [10], grapevine [11], rice husk [12], brewers spent grains [13,14], rice straw [15], olive stones [16], sorghum straws [17], flax shives, and sunflower bark [18,19], among others. However, while particleboards offer versatility, affordability, and ease of manufacture, concerns have emerged regarding the potential health risks associated with the use of synthetic adhesives [20]. The widespread use of urea-formaldehyde resins as adhesives in particleboard production has been linked to the emission of formaldehyde, a known carcinogen and respiratory irritant [3,21,22]. Despite efforts to mitigate these emissions through stringent regulations and the development of alternative adhesives with scavengers for formaldehyde [20], the health effects persist, posing significant challenges to both manufacturers and consumers [23,24]. This has acted as a significant driving force towards the design of more benign adhesive formulations from bioderived resources such as carbohydrates [25], proteins [23,24,26,27,28,29], and biophenols such as cardanol [30], tannins [31], and lignins [32,33].

Particleboards can also be manufactured without adhesives by subjecting lignocellulosic particles to specific humidity, heat, and pressure conditions [34,35]. Chemical compounds in lignocellulosic materials undergo chemical transformations during thermocompression in the presence of water, a process previously outlined in several studies [5,35,36,37]. Hot pressing induces various reactions, including dehydration, hydrolysis, and oxidation, primarily impacting hemicelluloses due to their lower thermal stability. These reactions induce the formation of new bonds serving as adhesives between particles [5,18,35,36]. Numerous studies have reported the feasibility of producing binderless boards from unconventional plant sources such as sugarcane bagasse [38], palm tree [39], kenaf [37], flax fiber [40], banana rachis [41], rice straw [42], totora [43], and sunflower [44], among others. As the proportions of lignin, hemicellulose, cellulose, proteins, and other substances vary depending on the plant source, it is essential to study the processing conditions in each case in order to obtain an adequate board [3,4,5,35,36,37,38,43,45].

Brewer’s spent grain (BSG) is a by-product of the beer-making process, consisting mainly of barley malt husks. It accounts for 85% of solid waste. BSG is classified as a lignocellulosic material and contains hemicelluloses (19–40%), cellulose (10–25%), lignin (11–27%), protein (14–30%), and polysaccharides such as starch, as well as less abundant amounts of non-polymeric components like phenolic compounds, alfa amino acids, fatty acids, vitamins, and minerals [46,47,48]. The high moisture content and chemical composition of BSG (mostly carbohydrates and proteins) make it susceptible to microbial spoilage and chemical deterioration by fungi, yeasts, and aerobic bacteria [13,49]. This limitation restricts its transportation from the brewery to short distances, although drying at the plant would facilitate its distribution beyond its production area. Various drying and preservation methods have been documented in the literature; however, care must be taken to avoid the effects of uncontrolled oven drying, autoclaving, and freezing, as these processes may alter the chemical composition or cause the loss of several components due to temperature or solubilization. Another option, but more costly and time-consuming, is freeze-drying [3,50,51,52,53,54,55].

The BSG recycling rate into value-added products is still low, and it is mainly used in animal feeding, human nutrition, the papermaking industry, biogas, and as a source of different chemical products and bioactive compounds, such as antioxidants [54,55,56,57,58,59,60,61,62]. However, the implementation of these developments on an industrial scale remains limited. Considering that BSG and wood have a similar chemical composition but differing amounts, converting BSG into high-value products such as particleboards is a sustainable alternative that could significantly reduce the volume of wasted BSG. In this regard, Klimek et al. [14] developed particleboards with varied BSG content (10, 20, 30, 50%) to replace wood particles using urea formaldehyde as adhesive. Although wood-BSG particleboards performed worse than those made entirely of wood, materials with 10% BSG still met general-purpose requirements. Barbu et al. [13] conducted a similar study but using a variety of synthetic adhesives, including urea-formaldehyde, melamine-formaldehyde, and polymeric diisocyanate. The use of 10% BSG resulted in boards that met the requirements for indoor use, regardless of the type of adhesive used.

In the current study, we propose, for the first time, the production of binderless particleboards based on BSG as the sole source of lignocellulosic material and natural adhesive. This study seeks to assess the impact of pretreatment methods and processing parameters on the mechanical and hygroscopic properties of these particleboards. In addition, the performance of binderless panels will be compared with a control panel using BSG combined with a phenolic adhesive.

## 2. Materials and Methods

### 2.1. Materials

BSG obtained from Pilsner malt was kindly provided by a local brewery in Mar del Plata (Argentina). Phenol-formaldehyde adhesive (PA) was provided by Industria Química Ltd.a. (Paraná, Brazil) (viscosity 200 Pa·s at 25 °C, density 1.21 g/cm^3^, solids content 54%, pH 12.3, and a gelation time of 9 min at 100 °C).

### 2.2. BSG Conditioning and Characterization

BSG was dried until a moisture content of 5 wt.%, in a circulating air oven (Memmert, Lucé, France) at 80 ± 2 °C and then immediately stored in hermetically sealed plastic containers at room temperature until its use. A fraction of the material was ground for one minute in a food processor (Drean, Argentina) to evaluate the effect of the particle size of the BSG on the properties of the particleboard. Both samples (BSG and ground BSG) were sieved to estimate the particle size distribution. The morphological structure of BSG was observed under a scanning electron microscope (FESEM)Crossbeam 350 (Carl Zeiss Microscopy GmbH, Oberkochen, Germany) with an accelerating voltage of 3 kv. Samples were chromium-coated in a vacuum sputter coater with a layer thickness in the nanometer range. The chemical composition of the BSG was determined according to the following methods and standards: lignin [63], α-cellulose [64], toluene-alcohol-soluble [65], ash [66], holocellulose [67], lipids [68], and proteins [69] using a conversion factor of 5.88 from the total nitrogen determination [70]. The hemicellulose content was estimated as the difference between holocellulose and cellulose [67].

### 2.3. Particleboard Preparation

#### 2.3.1. Binderless Boards

BSG-based boards were prepared with a target density and thickness of 1000 kg/m^3^ and 5 mm, respectively. Dry BSG was sprayed with distilled water at room temperature to achieve a moisture content of 35% *w*/*w*. Water was added in order to generate steam during the thermocompression process, which contributes to extracting and plasticizing the lignocellulosic compounds contained in the BSG. The boards were obtained by hot pressing using a hydraulic press (E.M.S. Argentina, Rosario, Argentina) and a steel mold (250 × 250 mm^2^). The mass of the BSG placed in the mold was calculated in order to obtain the target density and thickness. The boards were obtained by hot pressing using a hydraulic press (E.M.S. Argentina) and a steel mold (250 × 250 mm^2^). A pressure of 2.1 ± 0.1 MPa was chosen because it allowed the desired density and thickness of the board to be achieved. The time and temperature used are summarized in Table 1. The same process was carried out with BSG and ground BSG. These particleboards were designated BBx (BSG) and BBGx (ground BSG), where x is the pressing temperature (160, 170, and 180 °C). Three boards were made for each condition.

#### 2.3.2. PA Adhesive Boards (Control)

BSG and PA (PA = 10 wt.% solid of BSG) were blended for 10 min at room temperature in an orbital mixer (Silcook, Chengdu, China). Boards (control) were obtained by hot pressing, as described in Section 2.3.1. These particleboards were designated as PAB. The processing conditions are summarized in Table 1. A pressing time of 20 min was selected based on our previous reports [12]. Three boards were produced for characterization purposes.

### 2.4. Particleboard Evaluation

The particleboards were subjected to conditioning in a chamber set at 65 % relative humidity and 20 °C for 7 days prior to evaluation. The performance of the resulting particleboards was assessed following the standard procedures outlined in ASTM D 1037–93 [71].

The mechanical properties were assessed using an INSTRON-EMIC 2350 universal testing machine (São José dos Pinhais, Brazil). Flexural properties, such as modulus of rupture (MOR) and modulus of elasticity (MOE), were determined from rectangular strips measuring 50 mm × 200 mm × thickness (mm), employing a crosshead speed of 2.9 mm/min and a 130 mm span. Internal bond strength (IB) was measured using samples sized at 50 mm × 50 mm × thickness (mm) with a crosshead speed of 1.33 mm/min. Additionally, 50 mm × 50 mm × thickness (mm) samples were extracted from the tested flexure specimens to assess Density (D), Moisture Content (MC %), Thickness Swelling (TS), and Water Absorption (WA). Subsequently, samples designated for WA and TS underwent immersion in water at room temperature for durations of 2 and 24 h to analyze short- and long-term changes. Weight gain and thickness measurements were taken immediately after immersion. A minimum of 10 samples were evaluated for each property. The morphology of the boards was examined using FESEM under conditions identical to those described in Section 2.2.

### 2.5. Statistical Analysis

The statistical evaluation of experimental data was estimated using the one-way analysis of variance (ANOVA) along with Tukey’s tests at 95% confidence interval (α = 0.05).

## 3. Results and Discussion

### 3.1. BSG Characterization

The chemical composition and morphology of BSG particles play a pivotal role in the self-adhesion process of binderless boards. BSG is primarily composed of several layers. The outermost layer is the husk, which acts as a protective covering for the barley grain. It comprises cellulose, hemicellulose, lignin, and other polysaccharides, which contribute to its fibrous and tough nature [72]. Under the husk lies the aleurone layer, which is rich in proteins and lipids. Below these layers lies the endosperm, the starchy interior part of the barley grain, which contains carbohydrates and proteins. Residues from the endosperm and walls of the empty aleurone may persist in BSG, contingent upon the effectiveness of the mashing process [72,73]. Figure 1a displays the FESEM images of the BSG husk from the analyzed sample, representing the primary tissue type observed. The structure of the BSG husk appears porous, with cells exhibiting a substantial lumen volume and thin walls. Furthermore, the mashing process induced damage to the husk structure in certain areas, as depicted in detail in Figure 1b.

The chemical composition of BSG Pilsen is summarized in Table 2. The results obtained were similar to those reported [46,47,48].

The chemical constituents of BSG possess the potential for chemical transformations in the presence of water during hot pressing. This includes processes such as hydrolysis, dehydration, and oxidation of hemicelluloses, which can subsequently polymerize and contribute to particle-to-particle adhesion [18,74,75]. Additionally, the combination of moisture and heat can induce the degradation, softening, and plasticization of lignin, increasing the occurrence of chemical interactions and crosslinking reactions that can favor self-adhesion [3,37]. According to Hubbe et al. [3], self-adhesion is attained through effective mechanical contact between particles during the hot-pressing process. In this process, at least one of the natural components of the lignocellulosic material must be capable of deforming to increase the contact area, getting wetted, and establishing molecular-scale contact (such as covalent bonding or multiple secondary force interactions) between solid surfaces to ensure the structural integrity of the board. Bonding cannot occur in areas where there are spaces between the solid particles of the board. Consequently, the particle contact area becomes a significant variable in this process. To assess the impact of particle size on the properties of BSG boards, both raw and ground BSG were used in the manufacture of the binderless boards. As expected, ground BSG displayed a broader particle size distribution, with 65% of the particles falling within the range of 200–2380 µm, compared to 14% observed in raw BSG for the same range (Figure 2).

### 3.2. BSG-Based Particleboard

The processing conditions for the BSG particleboards were chosen based on our prior studies involving other substrates [12,28] and preliminary trials. Regardless of the applied pressing temperature, the pressure and pressing time for the binderless boards were maintained at 2.1 MPa and 15 min, respectively. The pressure level was selected because it facilitated achieving the targeted board density. The pressing temperature was determined using the thermal stability of BSG, as assessed by dynamic TGA (see Appendix A). The primary components of BSG (hemicelluloses, cellulose, and lignin) exhibited thermal stability at temperatures below 190 °C, as indicated by their respective temperature peaks and initial degradation temperature (IDT) (see Appendix A). Based on these findings, compression molding was conducted at temperatures of 180 °C, 170 °C, and 160 °C. The impact of pressing temperature was visually assessed by examining the surface appearance of the resulting boards (Figure 3). In the case of BB180 (Figure 3c), it is evident that the board experiences material loss and collapses before reaching the specified pressing time. This failure occurs when there is insufficient self-adhesion between particles (low internal resistance), coupled with a high rate of vapor generation, leading some BSG particles to explode in the mold. This, in turn, creates voids and depressurizes a portion of the board [76]. Once initiated, this process continues throughout the sample until the pressure or water content decreases sufficiently. Hence, it was determined that 180 °C was unsuitable for the intended purpose. Conversely, boards manufactured from both raw and ground BSG at 160 °C and 170 °C exhibit smooth surfaces devoid of flaws. The occurrence of browning in BSG-based particleboard could be attributed to Maillard reactions that may potentially occur during thermocompression. When proteins (17%) are present, reducing carbohydrates found in BSG may react to form oligomeric and polymeric melanoidins; even lignin (11%) with aldehyde groups can also participate. These compounds, melanoidins, are responsible for the brown color detected in the center of the board, where the temperature was slightly higher compared to the periphery [48]. This specific imperfection was taken into consideration during sample preparation for testing.

Table 3 summarizes the moisture content, apparent density, and thickness of the prepared BSG boards. No significant differences were observed in the equilibrium moisture content and apparent density of the particleboards (*p* < 0.05). The observed dispersions in thickness, given the targeted 5 mm thickness, are typical characteristics resulting from the compression process during board manufacturing.

The mechanical properties of the BSG boards are presented in Table 4. Increasing the pressing temperature enhanced the performance of the binderless boards in terms of flexural and internal bond properties. Among the BB boards, BB170 exhibited higher MOE, MOR, and IB values compared to BB160 (Table 4). Given that IB is associated with adhesion behavior, it can be hypothesized that processing at 170 °C may promote self-adhesion, as was reported for binderless boards based on other lignocellulosic materials [5,18,19,76]. The impact of particle size on the mechanical performance of the binderless board was also analyzed. BBG170 showed significantly higher MOR, MOE, and IB values compared to BB170 (*p* < 0.05), indicating that smaller particles contribute favorably to mechanical contact between particles, as previously described for boards derived from other lignocellulosic materials [3,4,77]. This behavior is likely attributed to the larger surface area and effective bonding between particles, involving the chemical compounds of BSG (hemicellulose, lignin, among others), as discussed previously [3,18,75]. The properties observed in BBG170 indicate that reducing the particle size of BSG and increasing the pressing temperature up to 170 °C facilitate chemical bonding and mechanical interlocking and improve the structural integrity of the boards. The MOE, MOR, and IB values of the BBG170 sample show similar results to those reported by [77] for sorghum-based binderless boards within the same density range.

On the other hand, the particleboard PAB bonded with PF (control) exhibited the best mechanical properties owing to the intrinsic chemical nature of the synthetic adhesive and its favorable interactions with the lignocellulosic substrate [12]. Phenolic adhesives can wet, spread, and penetrate the lumen and cell wall of lignocellulosic wastes [12,78,79] and form a strong bond between BSG particles through the formation of a covalently cross-linked polymeric network. Hydroxymethylphenol crosslinking occurs by heating to temperatures higher than 120 °C, forming methylene and methyl ether bridges by removing water molecules [80]. Additionally, the adhesive can permeate the cell wall of the BSG husk, which is possible through the diffusion of low-molecular-weight adhesive components [79]. Although PAB boards exhibited significantly improved mechanical properties compared to those without a binder, the mechanical results observed in the obtained boards were even lower than those achieved using other substrates such as wood [81] or rice husk [12] combined with synthetic adhesive. The low MOE values of BSG boards might be related to their low cellulose content (9.5%), compared to, for example, eucalyptus wood (approximately 40%) and other lignocellulosic wastes like rice husk (30%) [12,24,27,28]. A similar reduction in MOE values was also observed by Barbu [13] when substituting spruce wood with BSG in particleboard using different synthetic adhesives.

The response of the BB and BBG binderless particleboards to water immersion tests is presented in Table 5. The low WA and TS values after 2 and 24 h of immersion for BBG170 are consistent with the high IB value obtained for this sample due to the combination of high temperature and small particle size. A similar behavior was reported by Mahieu et al. [18] and Hashim et al. [82] for binderless boards made from sunflower and oil palm trunks, respectively. On the contrary, BB160 and BBG160 showed the highest WA and TS values after 2 h, and BB160 completely disintegrated after 24 h, in line with the lowest IB value obtained, reinforcing the hypothesis of poor interparticle adhesion of these boards. A temperature of 160 °C would not be sufficient to activate the chemical conversion of the hemicelluloses and lignin in BSG as well as the subsequent reaction of the resulting products.

On the other hand, PAB (control) showed the lowest values of WA and TS due to the high water resistance of phenolic adhesives, which makes them suitable for outdoor applications [12,81]. The PA resin has a low molecular weight and can penetrate the cell wall of BSG and react with the hydroxyl group of the lignocellulosic substrate. The formation of an insoluble polymeric network significantly improves the performance of boards under water [12,79].

As previously discussed, achieving adequate particle contact is a crucial factor for enhancing the mechanical performance and water resistance of binderless boards, facilitating the formation of secondary or covalent bonds among hemicelluloses, lignin, or their reaction products [3,75]. To assess this aspect, the morphology of the boards was analyzed by FESEM. Figure 4 presents FESEM images of BSG binderless boards obtained. BB160 and BBG160 boards (Figure 4a,b, respectively) exhibit a typical structure with discernible voids between the BSG particles. Notably, Figure 4c illustrates a distinct lack of particle contact, aligning with the lower IB values and higher WA and TS values recorded for these binderless boards (see Table 3 and Table 4). FESEM images of binderless boards processed at 170 °C (Figure 4d,e) revealed that both formulations demonstrated more compacted structures with reduced voids and pores compared to boards processed at 160 °C. This improvement is attributed to the higher temperature, which fosters chemical transformations and reactions among BSG components, thereby contributing to the self-adhesion process [18]. Comparing the FESEM images in Figure 4c (BB160) with Figure 4f (BBG170) reveals complete adhesion between the BSG layers in the sample pressed at 170 °C. This demonstrates the beneficial impact achieved by simultaneously increasing the pressing temperature and reducing the particle size. It is also evident that the porous cellular structure of the husk is retained despite the pressing process. The enhanced adhesion among particles at 170 °C significantly improved the mechanical strength and water resistance of the binderless boards [3,77].

PAB board (control) (Figure 5a,b) exhibits a densified structure and effective particle-to-particle contact due to the incorporation of PA. This morphology is consistent with the good mechanical and water performance of this panel. Furthermore, Figure 5b shows that the adhesive partially covers the cell lumen. Barbu et al. [13] observed a similar structure and behavior, highlighting that excessive adhesive consumption could occur due to lumen filling with adhesive.

## 4. Conclusions

BSG has demonstrated itself to be a sustainable alternative as the sole source of lignocellulosic material and adhesive to produce particleboard, with the additional benefit of being free from toxic volatile compounds such as formaldehyde, which enhances its eco-friendliness and safety.

Processing temperatures of 170 °C and small particle sizes (200 to 2380 µm) proved to be beneficial in facilitating particle-to-particle contact and the self-adhesion process. These conditions resulted in the production of binderless boards exhibiting improved flexural properties and enhanced water resistance.

Although BSG has a low cellulose content, which diminishes its inherent stiffness compared to other lignocellulosic residues or wood, the obtained boards can be effectively applied as interior or sheltered-use cladding panels or for ceilings where rigidity is not a critical factor.

Further research should be focused on combining BSG as the core material with natural fiber or fabric reinforcement to improve its flexural strength and rigidity. This approach could potentially lead to the development of reinforced panels with improved mechanical properties suitable for structural applications.

## Figures and Tables

**Figure 1 polymers-16-00059-f001:**
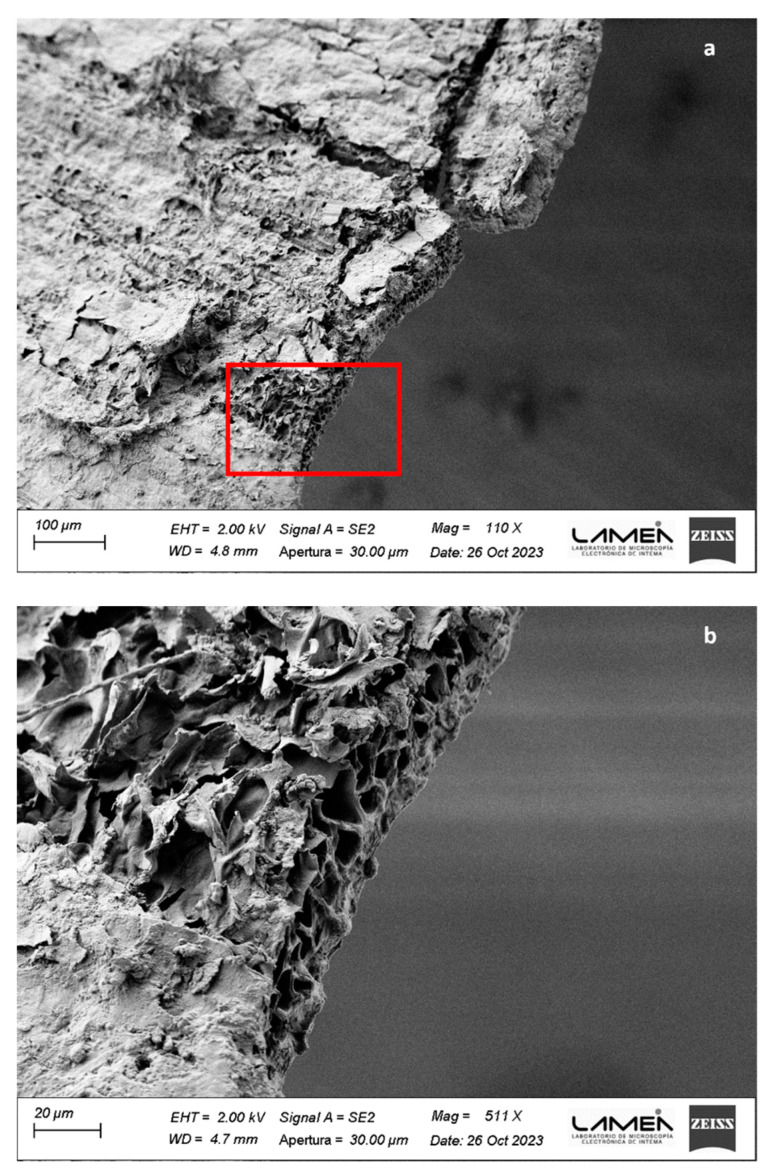
BSG husk image obtained by FESEM. (**a**) 110×; (**b**) detail at 511× of red square in (**a**).

**Figure 2 polymers-16-00059-f002:**
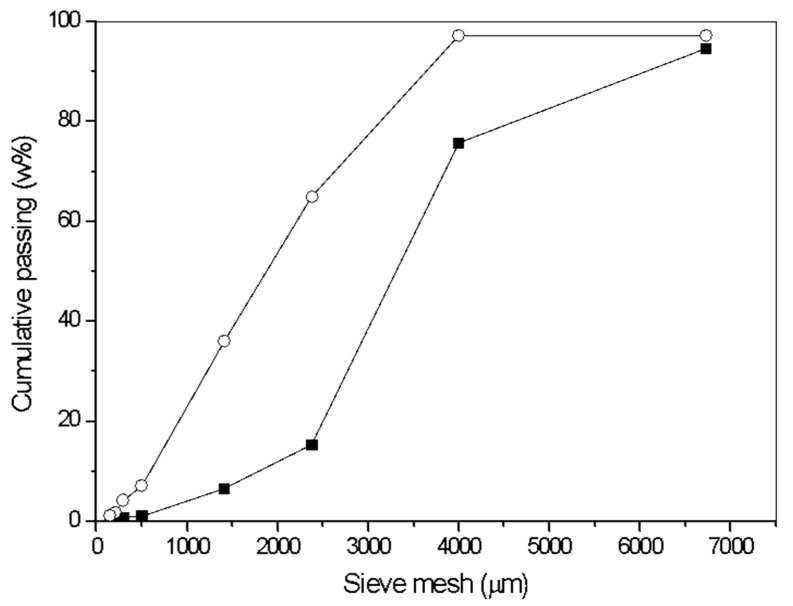
Particle size distribution of raw BSG (square) and ground BSG (circle).

**Figure 3 polymers-16-00059-f003:**
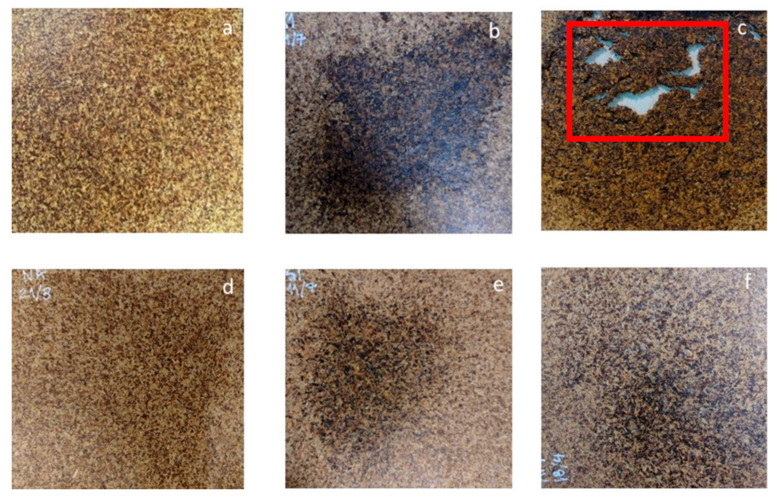
Particle boards Image (**a**) BB160, (**b**) BB170, (**c**) BB180, (**d**) BBG160, (**e**) BBG170, and (**f**) PAB. Red rectangle indicates the presence of macroholes in the board.

**Figure 4 polymers-16-00059-f004:**
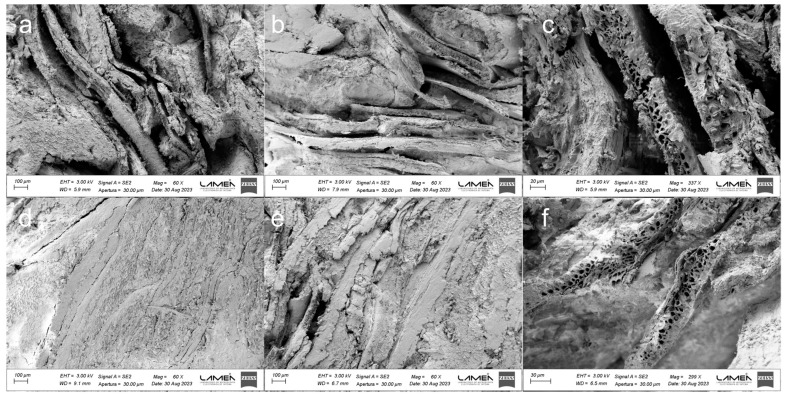
FESEM images of boards: (**a**) BB160 (60×), (**b**) BBG160 (60×), (**c**) BB160 (337×), (**d**) BB170 (60×), (**e**) BBG170 (60×), and (**f**) BBG170 (299×).

**Figure 5 polymers-16-00059-f005:**
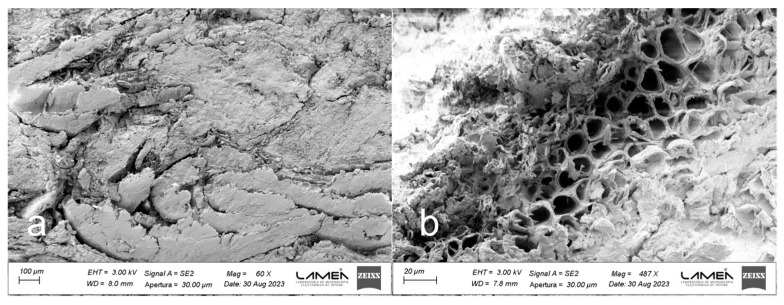
FESEM images of PAB boards (**a**) (60×), (**b**) (487×).

**Table 1 polymers-16-00059-t001:** Processing conditions of BBx, BBGx, and PAB. Pressure = 2.1 MPa.

Board	Pressing Temperature (°C)	Pressing Time (min)
BB160	160	15
BB170	170	15
BB180	180	15
BBG160	170	15
BBG170	170	15
PAB	150	20

**Table 2 polymers-16-00059-t002:** Chemical composition of BSG.

	Cellulose	Lignin	Hemicellulose	Protein	Ash	Fat
%	9.5	11.2	28.7	17.3	3.3	7.5

**Table 3 polymers-16-00059-t003:** Apparent density, thickness, and moisture content values of particleboard.

Board	Density (kg/m^3^)	Thickness (mm)	Moisture Content (%)
BB160	990 ± 50 a	5.27 ± 0.11 a	8.59 ± 0.26 a
BB170	1020 ± 41 a	5.00 ± 0.14 b	8.58 ± 0.38 a
BBG160	980 ± 60 a	5.20 ± 0.13 a	8.50 ± 0.45 a
BBG170	1040 ± 39 a	4.99 ± 0.13 b	8.52 ± 0.07 a
PAB	1050 ± 51 a	5.20 ± 0.10 a	8.23 ± 0.18 a

Mean values in the same column followed by different letters are significantly different (*p* < 0.05) by the Tukey’s Test.

**Table 4 polymers-16-00059-t004:** Results of flexural properties (MOR and MOE) and internal bond (IB) of boards.

Board	MOR (MPa)	MOE (GPa)	IB (MPa)
BB160	1.82 ± 0.25 a	0.46 ± 0.06 a	0.09 ± 0.03 a
BB170	2.43 ± 0.62 a	0.54 ± 0.23 a	0.15 ± 0.06 b
BBG160	2.06 ± 0.56 a	0.61 ± 0.13 b	0.09 ± 0.03 a
BBG170	4.14 ± 0.83 b	0.77 ± 0.15 c	0.23 ± 0.09 c
PAB	8.96 ± 0.72 c	1.09 ± 0.09 d	0.68 ± 0.13 d

Mean values in the same column followed by different letters are significantly different (*p* < 0.05) by the Tukey’s Test.

**Table 5 polymers-16-00059-t005:** Results of water absorption (WA) and thickness swelling of boards.

Board	WA 2 h (%)	WA 24 h (%)	TS 2 h (%)	TS 24 h (%)
BB160	153.2 ± 25.4 a	nd	126.5 ± 10.5 a	nd
BB170	83.6 ± 28.6 b	137.0 ± 4.2 a	45.4 ± 32.4 b	53.5 ± 13.4 a
BBG160	136.0 ± 25.0 a	171.6 ±11.2 b	80.2 ± 19.1 c	100.3 ± 10.2 a
BBG170	35.8 ± 9.5 b	79.8 ± 19.5 c	29.1 ± 8.7 b	52.3 ± 10.2 a
PAB	16.7 ± 2.4 c	35.4 ± 2.9 d	12.3 ± 2.2 d	34.5 ± 2.9 c

Mean values in the same column followed by different letters are significantly different (*p* < 0.05) by the Tukey’s Test. nd: not determined.

## Data Availability

The data presented in this study are available on request from the corresponding author.

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
