# Peer review of "Sustainable Particleboards Based on Brewer’s Spent Grains"

_polymers, 2023, doi:10.3390/polym16010059_

Round 1

Reviewer 1 Report

Comments and Suggestions for Authors

In this manuscript, the author reported synthesis and characterization of sustainable particleboard based on Brewers spent grains, which is a meaningful work. In addition, the whole manuscript was in a good organizing and writing, the following issues should be addressed before acceptance:

1. Extensive editing of English language required.

2. Line 21, “The impact of pressing temperature and particle size on properties such as...., however, only the impact of temperature was evaluated, the influence of particle size was not assessed in the manuscript.

3. Line 22, What is “internal bonding strength”? how can you determine the IB?

4. Table 1, why didnt you prepare the PAB particleboard at the same conditions with BB or BBG particleboard?

5. Line 148, what is “ground BSG”?

6. Line 171, how can you determine the moisture content?

7. Figure 1, (a) and (b) was not appeared within the Figure.

8. Line 202-203, the component of BSG should listed in a table.

9. Why does there are some white compounds appeared on the surface of Figure 3c? what is the component?

10. Table 4, what are the * stand for? the WA and TS of BB/BBG is far behind PAB, can you improve the WA and TS with change the processing parameters or particle size?

11. For the conclusions, data result should take into account.

Comments on the Quality of English Language

Extensive editing of English language required

Reviewer 2 Report

Comments and Suggestions for Authors

The experimental article “Sustainable particleboards based on Brewer’s Spent Grains” is devoted to research into the possibility of producing “particleboards” without a binder based on industrial waste from beer production, and specifically brewer’s grains (BSG). This type of waste such as BSG exists in every country in the world, but there are not many promising proposals for the disposal of BSG with implementation. The authors applied modern methods to study the properties of boards without a binder in comparison with boards with a commercial phenolic resin binder. According to many criteria (object, goal setting, methods, results based on the chemistry and materials science of polymers), this manuscript corresponds to the Polymers publication. The advantage of the article is the clear position of the authors to argue in favor of materials made from waste, justifying the environmental friendliness of the manufacturing method. The experimental article refers to 92 publications, but among them there is only one article from 2023, and under number 92, that is, used at the very end of the manuscript.

Recommendations for correction are given in a list.

1. Introduction. It is necessary to strengthen the rationale for the relevance of this topic using 2023 references.

2. Introduction. Explain the polymeric nature of BSG and briefly outline the problem of preserving this type of waste without drying.

3. Since hemicelluloses are necessary for the formation of BSG slabs, give a method for determining their content.

4. Lines 204-205. The sentence needs to be rephrased because TGA is not a method for determining hemicellulose content.

5. Line 251. Expand the sentence about the entity Maillard reaction, which leads to darkening of the surface of the slab.

6. Section 3.2. BSG based particleboard. It is necessary to briefly disclose the novelty of the results obtained for Polymers readers. The idea of producing boards without a binder has been tested on many types of lignocellulose. What fundamental patterns were discovered by the authors of this manuscript?

7. It is necessary to provide a forecast for the service life of products based on BSG in comparison with commercial ones.

8. Supplement the conclusions and abstract with a short sentence about the scientific novelty of the results obtained, taking into account the status of the publication.

Round 2

Reviewer 1 Report

Comments and Suggestions for Authors

All of the issues mentioned were resolved in detail.

Comments on the Quality of English Language

Moderate editing of English language required